# Using large ensembles to quantify the impact of sudden stratospheric warmings and their precursors on the North Atlantic Oscillation

Philip E. Bett[1], Adam A. Scaife[1,2], Steven C. Hardiman[1], Hazel E. Thornton[1], Xiaocen Shen[3,4], Lin Wang[3,4], Bo Pang[5]

[1]Met Office, FitzRoy Road, Exeter EX1 3PB, United Kingdom
[2]College of Engineering, Mathematics and Physical Sciences, University of Exeter, Exeter, United Kingdom
[3]Center for Monsoon System Research, Institute of Atmospheric Physics, Chinese Academy of Sciences, Beijing 100029, China
[4]College of Earth and Planetary Sciences, University of Chinese Academy of Sciences, Beijing 100049, China
[5]State Key Laboratory of Numerical Modeling for Atmospheric Sciences and Geophysical Fluid Dynamics (LASG), Institute of Atmospheric Physics, Chinese Academy of Sciences, Beijing, 100029, China

*Correspondence to*: Philip E. Bett (philip.bett@metoffice.gov.uk)

**Abstract.** Sudden stratospheric warming events (SSWs) are often followed by significant weather and climate impacts at the surface. By affecting the North Atlantic Oscillation (NAO), SSWs can lead to periods of extreme cold in parts of Europe and North America. Previous studies have used observations and free-running climate models to try to identify features of the atmosphere prior to an SSW that can determine the subsequent impact at the surface. However, the limited observational record makes it difficult to accurately quantify these relationships. Here, we instead use a large ensemble of seasonal hindcasts. We first test whether the hindcasts reproduce the observed characteristics of SSWs and their surface signature. We find that the simulations are statistically indistinguishable from the observations, in terms of the overall risk of an SSW per winter (56%), the frequency of SSWs with negative NAO responses (65%), the magnitude of the NAO responses, and the frequency of wavenumber-2 dominated SSWs (26%). We also assess the relationships between prior conditions and the NAO response in the 30 days following an SSW. We find that there is little information in the precursor state to guide differences in the subsequent NAO behaviour between one SSW and another, reflecting the substantial natural variability between SSW events. The strongest relationships with the NAO response are from pre-SSW sea level pressure anomalies over the polar cap, and from zonal wind anomalies in the lower stratosphere, both exhibiting correlations of around 0.3. The pre-SSW NAO has little bearing on its post-SSW state. The strength of the pre-SSW zonal wind anomalies at 10 hPa is also not significantly correlated with the NAO response. Finally, we find that the mean NAO response in the first 10 days following wave-2 dominated SSWs is much more strongly negative than in wave-1 cases. However, the subsequent response in days 11–30 is very similar regardless of the dominant wavenumber. In all cases, the composite mean responses are the result of very broad distributions from individual SSW events, necessitating a probabilistic analysis using large ensembles.

## 1 Introduction

Since their discovery (Scherhag, 1952), sudden stratospheric warmings (SSWs) have been recognised as some of the most dramatic events in the Earth's atmosphere. The strong winds of the winter stratospheric polar vortex (SPV) are disrupted, and even reversed, by the breaking of planetary-scale Rossby waves propagating upwards from the troposphere. The resulting descent of air over the pole causes adiabatic warming, with temperatures rising by several tens of kelvin over a matter of days (e.g. Baldwin et al., 2021 and references therein). The zonal mean signature of this disruption propagates downwards (Kodera, 1995), leading in some cases to impacts at the surface (Baldwin and Dunkerton, 1999, 2001; Christiansen, 2001), enhanced by eddy feedbacks in a way that is only partly understood (Kidston et al., 2015; Kunz and Greatbatch, 2013). In the Northern Hemisphere, the changes in the tropospheric circulation are often characterised in terms of negative anomalies in the Arctic Oscillation (AO) and the North Atlantic Oscillation (NAO), reflecting an equatorward shift of the jet stream and storm track (Baldwin and Dunkerton, 2001). This leads to corresponding impacts on the weather (e.g. Butler et al., 2017; King et al., 2019), with warm anomalies in eastern Canada, and cold anomalies in the eastern United States and across northern Eurasia. The northern European regions of Scandinavia and the British Isles experience reduced precipitation, while central and southern Europe experience wetter than average conditions. In East Asia, SSWs can affect the East Asian Winter Monsoon (Deng et al., 2008): Although China on average tends to experience milder conditions following an SSW (Lim et al., 2019), East Asia in general can see an increased risk of extreme cold air outbreaks (Huang et al., 2021; Kolstad et al., 2010; Song et al., 2015). In general, SSWs are linked to extremes in surface climate (e.g. Domeisen and Butler, 2020; Huang et al., 2021), with potentially severe impacts on human health and wellbeing (e.g. Charlton-Perez et al., 2021).

Although individual SSWs themselves are predictable in a deterministic sense on timescales of one to two weeks (Taguchi, 2016), their prolonged disruption of the stratosphere and impact at the surface means that the occurrence of an SSW can increase the predictability of the subsequent surface climate to one or two months (Scaife et al., 2022; Sigmond et al., 2013). The predictability of SSW events is improved in models with a better-resolved stratosphere (Marshall and Scaife, 2010), although this is a necessary but not sufficient condition for good representation of SSWs: Chávez et al. (2022) for example showed that using a coupled ocean model had more impact than vertical resolution. The overall winter risk of an SSW occurring can also be predicted probabilistically with some skill at lead times of several months (Scaife et al., 2016), although the skill is dependent on other concurrent climate features such as El Niño.

SSWs have been observed to occur approximately 6 times per decade in the Northern Hemisphere (Bancalá et al., 2012; Charlton and Polvani, 2007), and only about 2/3 of SSWs are followed by effects at the surface as described above (e.g. White et al., 2019). A number of studies have used observation-based data (reanalyses) to investigate precursors of SSWs in surface climate features, as well as in the wave driving from the troposphere and in characteristics of the vortex itself (e.g. Bao et al., 2017; Cohen and Jones, 2011; Domeisen et al., 2020; Martius et al., 2009; Mitchell et al., 2013; Nakagawa and Yamazaki, 2006; Polvani and Waugh, 2004; Seviour et al., 2013; Shen et al., 2020; Xu et al., 2022). However, the variability seen between

different SSWs, and in the climate conditions in which they occur, coupled with their relatively low frequency and the limited observational record (only a handful of decades), has made it difficult to make definitive statements on what, if any, preconditioning causes stronger surface impacts following one SSW than another.

Recognising the limited observational sample, some studies have sought to increase their sample sizes by using free-running climate models (e.g. Garfinkel et al., 2010; Karpechko et al., 2017; Kolstad et al., 2010; Kolstad and Charlton-Perez, 2011; Maycock and Hitchcock, 2015), performing dedicated model experiments (e.g. de la Cámara et al., 2017; White et al., 2021), or using "ensembles of opportunity" from climate model experiments designed for other studies (e.g. White et al., 2019). Although helpful, these approaches are not without problems. The climate model runs used might not be designed to simulate the climate over the same period as the observations, and might be subject to biases or trends that grow over the course of the runs. Different models will be subject to different biases, and this can make it difficult to interpret results in terms of uncertainty. Hall et al. (2022) demonstrated that although models from the recent Coupled Model Intercomparison Project 6 (CMIP6) ensemble performed well in terms of their responses to SSWs, they exhibited different tropospheric precursors compared to observations. Tyrrell et al. (2022) showed that the weak SPV in their model experiments resulted in too many SSWs, and showed that a nudging bias correction method could improve this. However, the mean sea level pressure response to SSWs in their model was neither biased, nor affected by their bias correction. An alternative approach is to bootstrap the observational sample itself (e.g. Oehrlein et al., 2021), which allows the uncertainty in the observed sample to be estimated. Nevertheless, the results do not necessarily span the full range of possible present-day climate variability due to the inherent limitations of the observed sample size.

As a result of these limitations, existing studies do not always agree. For example, Mitchell et al. (2013) and Seviour et al. (2013) found that whether an SSW is characterised by the vortex splitting or simply being displaced has a significant impact on the subsequent surface response. In contrast, Charlton and Polvani (2007), Cohen and Jones (2011), and Maycock and Hitchcock (2015) found that the differences were small, subject to sampling variability, and not robust to changes in methodology or data. The uncertainties brought about by the limited observational sample and compounded by possible errors in climate models have meant that there is still no real consensus on these questions.

Instead, in this study, we use a large ensemble of initialised climate simulations, produced as seasonal hindcasts. In contrast to free-running climate models, these initialised simulations are more closely constrained to describe variability within the recent observed climate, while also providing a much larger data set. This follows the UNSEEN approach (UNprecedented Simulated Extremes using ENsembles, Thompson et al., 2017), in which a large ensemble of initialised hindcasts is used to greatly increase sample sizes, to quantify the probability of plausible but unobserved climate states (e.g. van den Brink et al., 2004, 2005; Brunner and Slater, 2022; Kay et al., 2020; Kelder et al., 2020; Kent et al., 2017). In our case, we are not quantifying rare weather extremes, but are nevertheless interested in the likelihood of particular climate events and their responses. This approach has already been applied to SSWs in some cases, e.g. for assessing the chance of southern hemisphere sudden

warmings and associated risk of extreme hot/dry conditions in austral subtropical continents (Wang et al., 2020), looking at the subseasonal impacts on the Arctic Oscillation (Spaeth and Birner, 2022), examining the impact of SSW timing on the resulting surface weather (Monnin et al., 2022), and the relationship between the impact of strong and weak vortex events and sea surface temperatures (Kolstad et al., 2022). Here, we will focus on quantifying how features of the pre-SSW climate affect the probability of the negative NAO conditions that drive the weather response at the surface.

The model and observational data we use are detailed in Sect. 2, together with descriptions of how we characterise SSWs and their responses. In Sect. 3 we demonstrate the accuracy of our model data in representing SSWs and their surface impacts, in comparison to the observations and their sampling uncertainty. Sect. 4 examines what determines the NAO response to SSWs, by considering precursors at the surface and in the stratosphere. We discuss and summarise our results in Sect. 5.

## 2 Data and methods

### 2.1 Climate model hindcast and observation-based data

We use hindcast data from the GloSea5 seasonal forecast system (MacLachlan et al., 2015), which is based on the HadGEM3-GC2 coupled climate model (Williams et al., 2015). The model has an atmospheric grid spacing of $0.833°$ longitude and $0.556°$ latitude, and 85 vertical levels extending to a height of 85 km. GloSea5 has been shown to have a good representation of the total variance of the NAO (Scaife et al., 2014). The hindcasts cover 23 winters, 1993/94 to 2015/16. Daily data for an extended winter period (December–March, DJFM) are used from predictions initialised on three dates centred on early November (25th Oct, 1st Nov, 9th Nov). These initialisation dates provide a balance between ensuring that the model has enough time for spin-up, while also incorporating information about the seasonal climate prior to each winter. We have 14 ensemble members available per initialisation date. This therefore yields an ensemble of $14 \times 3 = 42$ members per winter, leading to an overall sample of $42 \times 23 = 966$ winters altogether. The observational record we use is from ERA5 (Bell et al., 2021; Hersbach et al., 2020), covering 72 winters from 1950/51 to 2021/22. This has a grid spacing of $0.25° \times 0.25°$, and 37 vertical levels. Daily means are calculated from the hourly data available for download.

In order to make fair comparisons between the model data and ERA5, we resample the model data into a series of 1000 ensembles, each with the same number of winters as the observations. We can then test whether the single observational sample of 72 years is consistent with the distribution of possible 72-year samples seen in the model hindcasts, allowing us to account for sampling uncertainty due to climate variability. In each of the 1000 resamples, an ensemble member is randomly picked from the hindcast (with replacement), either from the same year (for the 23 years within the hindcast period), or from across the whole sample of 966 winters (for the remaining 49 years).

## 2.2 Methods

We use daily mean pressure at mean sea level (PMSL), zonal wind, and geopotential height (GPH) data. To calculate anomalies, we create daily climatologies for each variable, smoothed using a Gaussian filter with a width (standard deviation) of 10 days. The daily climatologies are calculated separately, but in the same way, for both reanalysis and model hindcast data; the only difference being that they are based on 72 winters for the reanalysis, and 966 winters for the model data. We have checked that there are no trends in these variables that need removing, in either the reanalysis or model data.

We define an SSW event simply as the first day in a DJF period (ignoring any leap days) when the zonal mean zonal wind at 60° N and 10 hPa goes below zero (following Charlton and Polvani, 2007). We therefore measure zero or one SSW per winter. Defining SSWs only in DJF, rather than DJFM, helps separate genuine SSW events from the final warming at the end of a winter, as well as ensuring that we always have at least 30 days of data after every SSW to assess their subsequent impact. For some analyses, we will also require a fixed number of days before each SSW, to assess the impact of precursor climate features.

Typically this will be 10 days, which restricts our sample of SSWs in these cases to those occurring on or after 11[th] December, as our data sets start on 1[st] December. The 30-day post-SSW period and 10-day pre-SSW periods were chosen after examining the composite mean time series of data before and after the SSW events, discussed in the results below, although other time periods were also tested.

We focus on the impact of SSWs on the North Atlantic Oscillation (NAO). We define an NAO anomaly index as the difference
between PMSL anomalies in two boxes, one near the Azores (28° W to 20° W, 36° N to 40° N) minus one over Iceland (25° W to 16° W, 63° N to 70° N), following Dunstone et al. (2016). In addition to the NAO anomalies before and after SSWs, we also consider the "climatological" NAO anomalies, i.e. independent of whether or not there is an ongoing SSW. For this we calculate the mean NAO anomalies in 5,000 random 30d periods that start in DJF (i.e. mirroring our SSW definition), from across our 966 model winters. The proportion of negative NAO anomalies can then be calculated (43%).

We characterise our SSW events according to the dominant zonal wavenumber in the vortex (cf. Martius et al., 2009; Nakagawa and Yamazaki, 2006). We calculate the amplitudes of zonal waves 1 and 2 ($A_1$ and $A_2$ respectively), from the Fourier transform of the daily mean eddy geopotential height at 60° N and 50 hPa (results using 10 hPa are similar). We focus on zonal wavenumbers 1 and 2, as there is very little contribution from higher wavenumbers at these altitudes, and consider SSW events with $A_2 > A_1$ on the date of the SSW to be wave-2 dominated, and those with $A_2 < A_1$ to be wave-1 dominated (results using
the mean amplitudes over the 10 days pre-SSW are similar). SSWs dominated by wave 1 will tend to correspond more to "displacement" events, and wave-2 dominated events will involve a split in the vortex. However, all events will involve a mixture of different wavenumbers, and there is no direct correspondence between our wave-1/wave-2 dominated classification and a displacement/split classification based on vortex geometry (Martius et al., 2009; Seviour et al., 2013; White et al., 2019).

Finally, we calculate 95% confidence intervals on correlations using a Fisher Z transformation (e.g. Wilks, 2020), and on
proportions/frequencies using a Wilson interval (e.g. Brown et al., 2001). We use a standard binomial test to compare a sample

proportion to a binomial distribution, the standard Gaussian approach for binomial statistics to test if two sample proportions are significantly different to each other, and the standard student's *t* test to assess if two means are significantly different. These tests are all performed at the 5% level.

### 3 Does the seasonal forecast system represent stratosphere–troposphere coupling accurately?

Histograms showing the frequency of SSWs and their NAO responses are shown in Figure 1, with the overall SSW frequency shown in Figure 1(a). In the model, 545 winters out of 966 have at least one SSW, i.e. 56% (with a 95% confidence interval of 53%–60%, and significantly different to 50% using a binomial test). This compares with 34 winters out of 72 in ERA5 (47%, not significantly different to 50%). The 95% range from our model resamples is 41%–72%, covering the observed value, and giving an indication of its larger uncertainty due to the more limited sample size. The observed frequency is therefore

statistically indistinguishable from the more robust estimate from the model hindcast.[1]

---

[1] We have also tested that resampling the model can reproduce the SSW frequencies seen in each 23-year period in the observations, i.e. that despite the difference between the hindcast period (1994–2016) and ERA5 period (1951–2022), our model is nevertheless able to reproduce the diversity of situations seen in the observations.

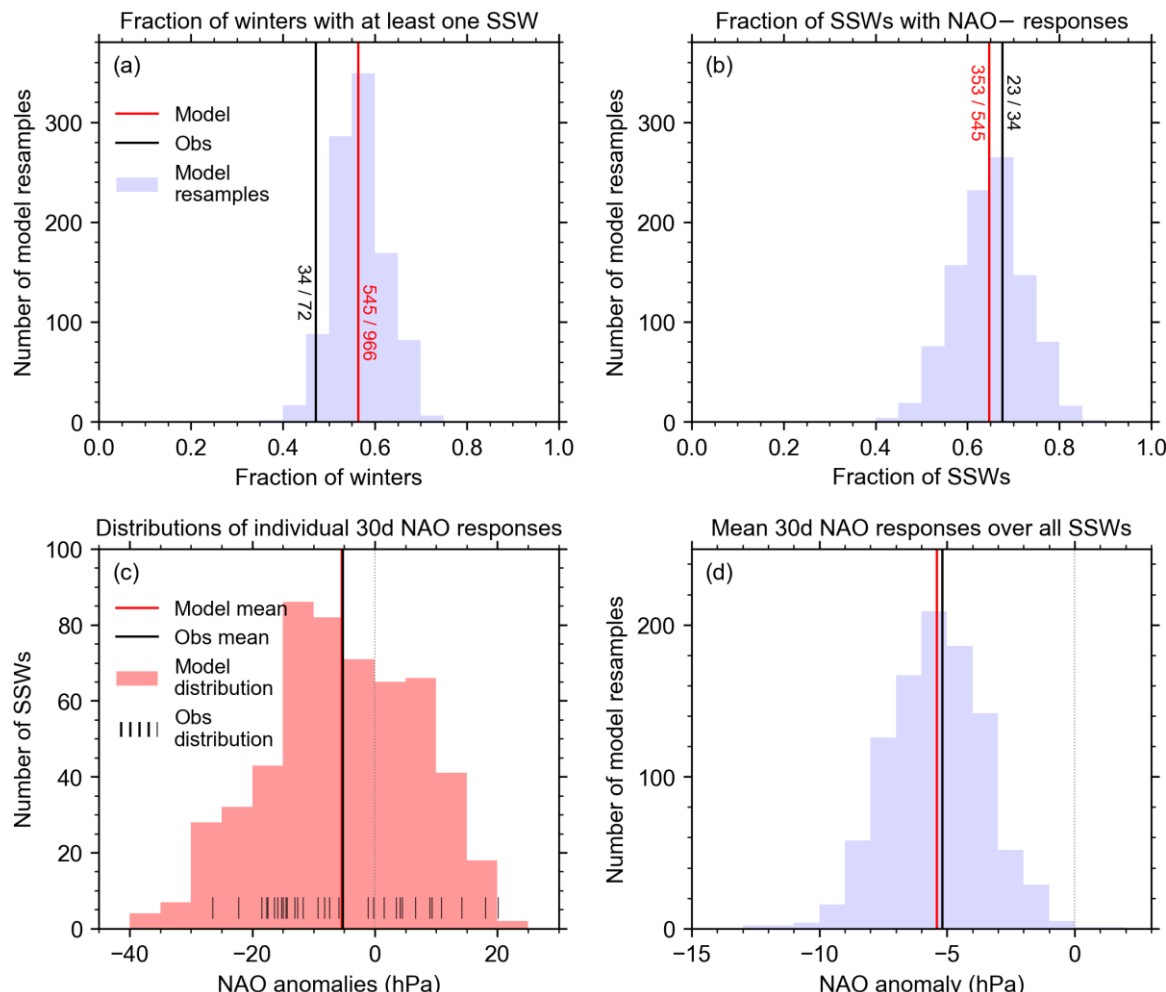

**Figure 1. The frequency of SSWs and their NAO responses. Panels show (a) the fraction of winters with at least one SSW in DJF; (b) the fraction of those SSWs with negative NAO responses, based on the mean NAO anomaly in the 30d following the SSW; (c) the distribution of the post-SSW NAO anomalies, over all SSWs; and (d) the mean of those post-SSW anomalies, over all SSWs. All panels use the same colouring: Solid vertical lines show the single values from the observations (black) and the model (red), and blue histograms show the distributions of such values over the 1000 72-year model resamples. In panel (c) the red histogram shows the distribution over the 545 SSWs in the model, and the black ticks at the bottom of the plot show the corresponding observed distribution. Note that the solid vertical lines in panels (c) and (d) show the same means, and the faint dotted vertical lines indicate zero.**

The NAO responses are examined in terms of the mean NAO anomaly in the 30 days following an SSW. The frequency of a negative NAO response is shown in Figure 1(b) to be about 2/3, consistent with previous results (e.g. Domeisen, 2019; Karpechko et al., 2017; Sigmond et al., 2013). The model has 353 SSWs that are followed by a 30d mean negative NAO, out of 545 (65%, with 95% confidence interval of 61%–69%). The observed proportion is very similar, at 23 SSWs out of 34

(68%). However, the model resamples suggest a much wider range of NAO-negative frequencies was possible from a 72-year sample like the observations, with a central (95%) range of 50% to 79%.

These frequencies can also be compared to the probability of 30d negative NAO conditions *without* necessarily following an SSW. If we examine the NAO anomalies from 5,000 random 30d periods that start in DJF, then the chance of them being negative is 43% in the model, with a 95% confidence interval of 42%–45%, and 46% in the observations, with a 95% confidence interval of 44%–47%. These values are below 50% because the distribution of NAO anomalies is negatively skewed. Our result of a 65% probability of a negative NAO response therefore represents a significant increase in the chance of a negative NAO month following an SSW.

The distribution of individual post-SSW NAO responses is shown in Figure 1(c), and the distribution of mean NAO responses (averaged over all SSWs) across the hindcast resamples is shown in Figure 1(d). There is a very broad range of possible NAO responses in the model, including positive as well as negative outcomes, although the distribution is strongly shifted towards negative NAO conditions. The observations and the model span very similar ranges, and have very similar mean responses (which are not significantly different). We can therefore say that the NAO response to SSWs in our model is indistinguishable from the observations.

We have performed a similar analysis using the Arctic Oscillation (AO, not shown), defined as the mean PMSL anomalies in the region 40° N to 60° N minus that from 60° N to the pole, and we find similar results: for the distributions and mean AO responses, and the frequency of negative AO responses, the results for the model and observations are statistically indistinguishable from each other.

Composite mean vertical profiles of the zonal mean zonal wind anomalies at 60° N are shown in Figure 2, following the similar plots of Baldwin and Dunkerton (2001). These illustrate the mean vertical progression of anomalies in the zonal mean circulation following an SSW, and again demonstrate the impact of the different sample sizes on the results. The observational composite is clearly "noisier" than the model data due to the limited number of events. However, the observations are broadly consistent with the model resamples: the only large areas where the observations lie outside the 95% range of the resamples (pink cross-hatching) are at higher levels in the stratosphere (above 20 hPa), immediately before the SSWs and at 35–70d afterwards; note that at these longer lags, there are progressively fewer observed events in our sample.

Both model and observations exhibit a near-surface easterly anomaly for at least the first 30 days following the SSW. This motivates our use of the 30d post-SSW period for assessing the NAO responses. There is also a consistent signal in the model and observations for easterly anomalies through the depth of stratosphere and troposphere from about 10 days before the SSW. We therefore use this 10d pre-SSW period later when considering precursors to SSW events.

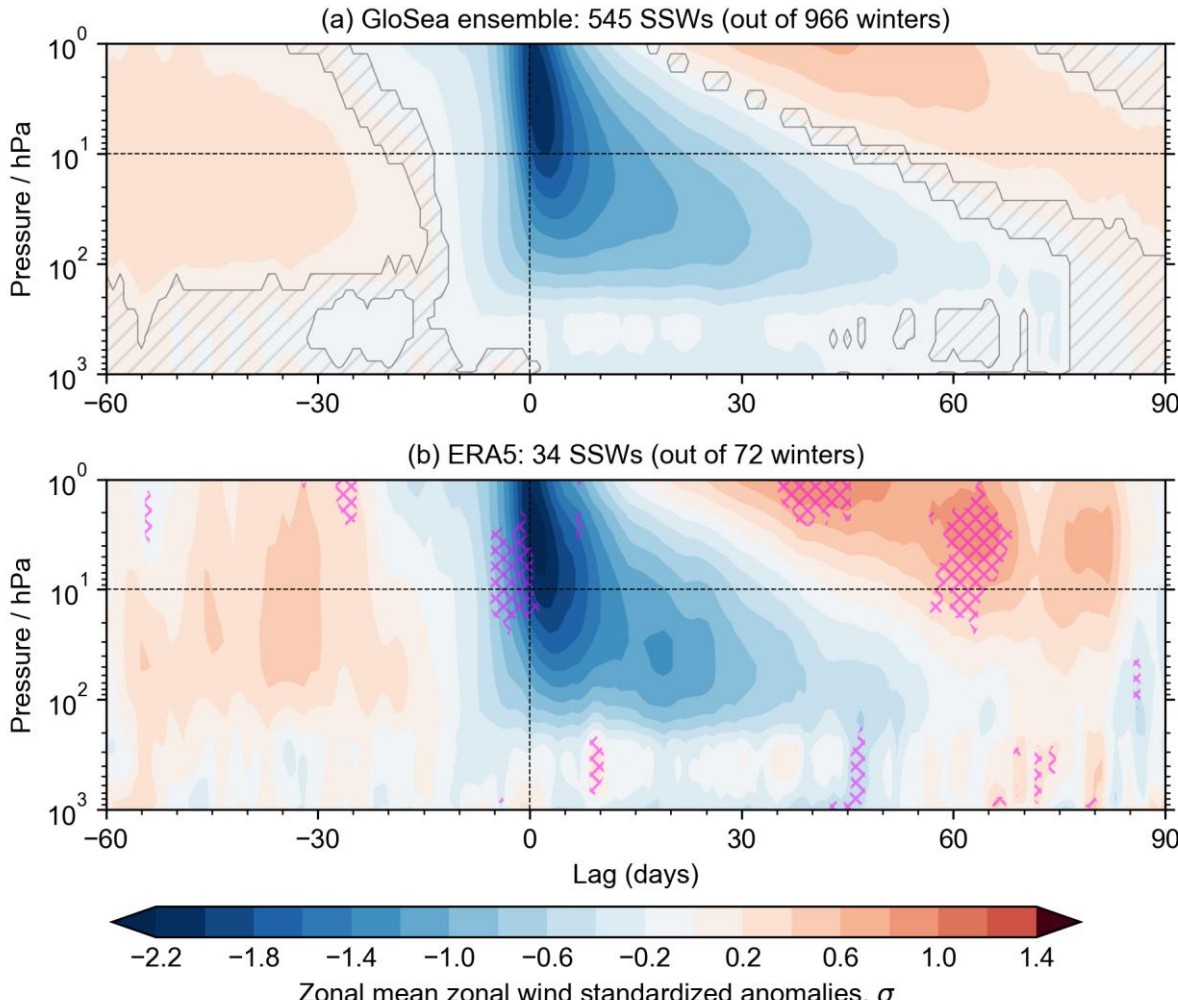

**Figure 2. Development of zonal mean zonal wind anomalies associated with SSWs in (a) models and (b) observations. Both panels show composite mean standardized anomalies in the 60° N zonal mean zonal wind profiles, with each contributing winter centred at their SSW date (lag = 0). Only dates in DJFM contribute; the composite comprises fewer SSWs at large lags/leads. In the model panel (a), areas are hatched where the anomalies are indistinguishable from zero. In the observation panel (b), pink cross-hatching marks where the anomalies exceed the 95% range of the hindcast resamples.**

Finally for this section, Figure 3 shows the PMSL composite means over the 30 days following SSWs. The mean negative NAO response shown in Figure 1(d) can be seen clearly here, and although the response is nominally weaker in the model, there is no statistically significant difference between model and observations in the North Atlantic or Arctic regions: the observations fall within the range expected from the model resamples.

Two regions in the observations, one from eastern Europe to northeast Africa, and another from the central United States to Central America, do however lie outside the model resamples' 95% range (purple cross-hatching), and similar areas show significant differences with the model mean (panel c). Although these responses could merit further investigation, we do not

expect them to affect our results here. Figure 3 is broadly consistent with previous results from models (e.g. Charlton and Polvani, 2007) and reanalysis (e.g. Butler et al., 2017).

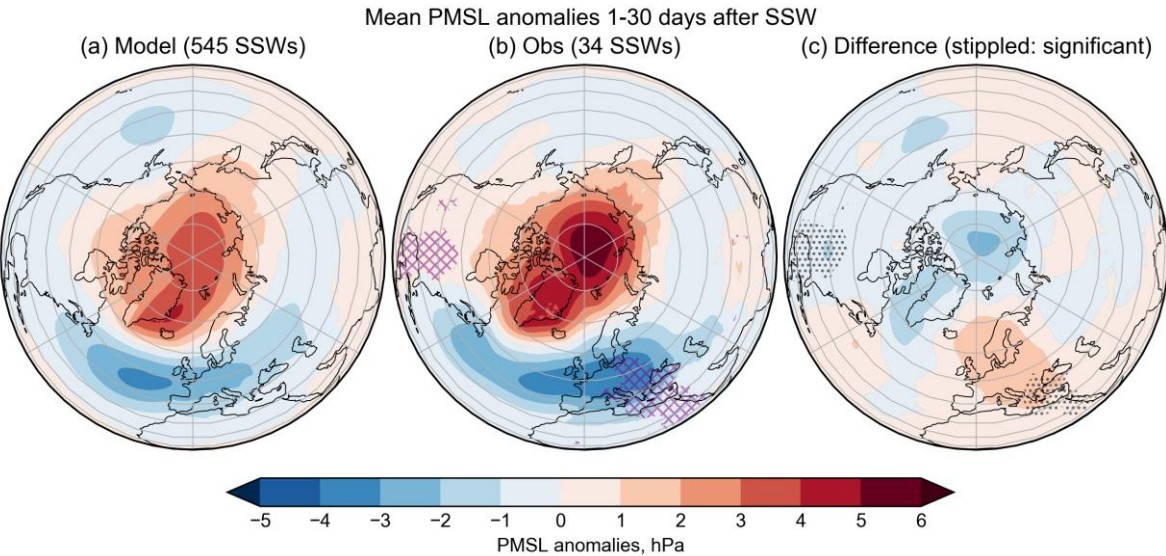

Figure 3. Surface responses to SSW events. The left and centre panels show the composite mean PMSL fields for (a) the model and (b) observations, in the 30d after an SSW. In the observational panel, regions where the data exceeds the 95% range of composites from the hindcast resamples are marked with purple cross hatching. The right-hand panel (c) shows the difference between the mean fields (model minus observations), with stippling indicating where the differences are statistically significant.

We can therefore conclude that the model is statistically indistinguishable from the observations for the features that we focus on in this study: the frequency of SSWs, the distribution and strength of post-SSW 30-day mean NAO responses, and the frequency of negative NAO responses.

## 4 What affects the NAO response to SSWs?

Having established the realism of our large model ensemble, we now assess atmospheric features that might have a robust impact on the NAO response.

### 4.1 Tropospheric precursors of negative NAO responses

Following Figure 2, we use a 10d pre-SSW period to examine the impact of surface precursors on the 30-day post-SSW NAO response. As discussed in section 2.2, this results in a slightly smaller sample of SSWs: 507 from the model rather than 545, and 32 from ERA5 rather than 34. We first note that the pre-SSW NAO itself is not a strong precursor of a negative NAO response. The correlation between the pre-SSW and post-SSW NAO anomalies in the model is 0.19, which is statistically significant but very small. This is consistent with Christiansen (2005), who showed a similar result for the AO, and with

Domeisen et al. (2020), who used a weather regimes approach to examine tropospheric evolution around SSWs. Another way of assessing the impact of a potential precursor is to use its terciles (for example) to select SSW events, and compare the resulting likelihoods of negative NAO responses in those subsets. If we pick SSWs according to the upper and lower terciles of the pre-SSW NAO, then the probability of a post-SSW negative NAO changes from 65% for the full sample (as in Figure 1b), to 66% and 72% respectively; these are not significantly different to each other.

We can identify other regions that might act as surface-level precursors of negative NAO responses by mapping the correlation between pre-SSW PMSL and post-SSW NAO anomalies (Figure 4). The weak correlations seen in the model data emphasize the very high degree of variability from one SSW to another, and thus provide important context for how we interpret other results. Large areas of PMSL over the polar cap, the subtropical North Atlantic, and North Pacific show statistically significant correlations with the post-SSW NAO response. However, most of these regions are not statistically significant, or not as 250 strongly correlated, in the observational map. Rather than the whole polar region, the observations show an area of apparently stronger correlation over Siberia, with peaks over the Ural region and the Russian Far East. If we consider area-averaged PMSL anomalies, the correlation of the Ural region pre-SSW (as marked on the map) with the post-SSW NAO in the model is small (-0.21, although significantly different to zero). If we select SSW events according to the upper and lower terciles of the pre-SSW Ural PMSL anomalies, we find this changes the probability of a negative NAO response in the model from 65% 255 (as in Figure 1b) to 74% and 56% respectively, which are significantly different from each other.

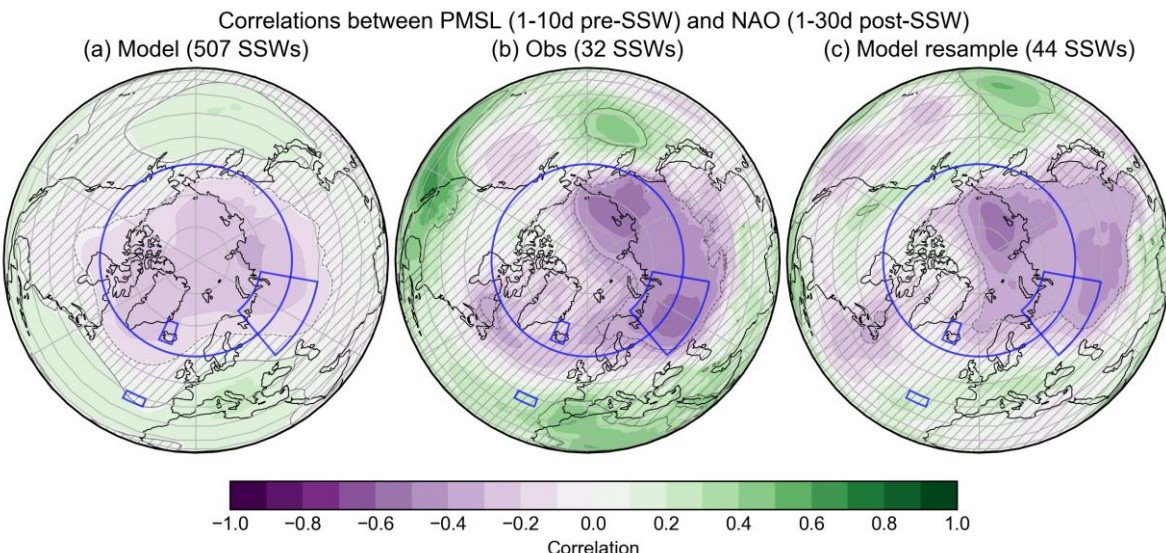

Correlations between PMSL (1-10d pre-SSW) and NAO (1-30d post-SSW)
(a) Model (507 SSWs)  (b) Obs (32 SSWs)  (c) Model resample (44 SSWs)

−1.0  −0.8  −0.6  −0.4  −0.2  0.0  0.2  0.4  0.6  0.8  1.0
Correlation

**Figure 4. Correlation of surface precursors with the NAO response to SSWs. The left and centre panels show the correlations between the 10d pre-SSW PMSL fields and the 30d post-SSW NAO anomalies for (a) the model and (b) observations. The right-hand panel (c) shows the correlation map for a particular case from the hindcast resamples, chosen as the sample whose correlation**
**map has the highest spatial correlation with the observed correlation map for points north of 40° N (0.83). In each panel, the 95% confidence intervals are marked with a contour, and areas where the correlation is indistinguishable from zero are hatched. Regions used in our analysis (the NAO boxes, a Ural region, and the Polar cap) are outlined in blue.**

If we instead pick a region covering the Polar Cap (PMSL anomalies averaged north of 60° N), we find that the post-SSW NAO-negative probabilities are more strongly differentiated into 78% and 53% for the upper and lower thirds[2] of the pre-SSW

PMSL anomalies respectively, and the correlation is stronger too, at -0.34. High pressure or blocking, approximately over Eurasia (and often the polar cap), has often been found to be a precursor to SSWs in general (e.g. Cohen and Jones, 2011; Kolstad et al., 2010; Kolstad and Charlton-Perez, 2011; White et al., 2019). However, our results extend this by quantifying the impact of these regions as precursors of post-SSW negative NAO conditions.

Our results demonstrate that just using the observations to select and test the impact of possible SSW precursors might result

in a sub-optimal choice of areas. Using the model data allows more statistically robust signals to emerge from the noise, allowing us to be more confident in the results. Figure 4 also shows the correlation map from the 72-winter model resample that has the greatest *spatial* correlation with the observed correlation map north of 40° N, highlighting that the model can produce similar relationships to those seen in the observations; the differences with the full model map are therefore largely due to better sampling.

**4.2 Zonal wind precursors**

We extend the above approach to examine how the post-SSW NAO response changes when we subselect SSWs according to the 60° N zonal mean zonal wind (ZMZW) anomalies at different heights, and different lead times, prior to the SSW. The impact on the probability of a negative NAO response is shown in Figure 5, in terms of the difference between the probabilities from picking lower and upper thirds of the ZMZW on each pressure level; a positive difference indicates an NAO-negative

response is more likely for the SSWs preceded by more easterly ZMZW anomalies at a given height and lead-time. For the immediate 10-day pre-SSW period (rightmost column in Figure 5), as used in the previous section, the ZMZW in the lower stratosphere and troposphere can clearly affect the probability of a negative NAO response: if we pick SSWs in the upper and lower thirds of the ZMZW at 100 hPa, then the subsequent NAO-negative probabilities are 56% and 73% respectively. The correlation of the ZMZW at 100 hPa with the post-SSW NAO is 0.30 (not shown), which is also statistically significant.

In contrast, if we focus on the 10 hPa level (which we use to define whether or not there is an SSW), there is no significant relationship between the 10d pre-SSW values and the post-SSW NAO: selecting by terciles only changes the NAO-negative probabilities to 61% and 67% (not significantly different), and the correlation with the NAO is 0.09 (also not significant).

Figure 5 also shows how these features vary as we examine earlier pre-SSW periods. Although there are some signs that the 10 hPa ZMZW can be a significant precursor around 2–3 weeks before the SSW, the stronger signals remain robustly in the

lower stratosphere: e.g. terciles of the 21–30d pre-SSW ZMZW at 100 hPa can still separate the post-SSW negative NAO

---

[2] Although our choice of separating the distribution at its terciles is arbitrary, our results are qualitatively unchanged if we select by different quantiles. If we use the outer quintiles of the pre-SSW polar cap PMSL (resulting in smaller samples), the probabilities of negative NAO post-SSW are instead 82% and 49%.

probabilities into 61% and 75%. The relationship between the pre-SSW lower stratosphere and the post-SSW NAO remains significant for 10-day periods as early as 3–4 weeks before the SSW. Therefore selecting SSWs according to the zonal mean zonal winds in the lower stratosphere is a robust way of selecting events that are more/less likely to result in a negative NAO response.

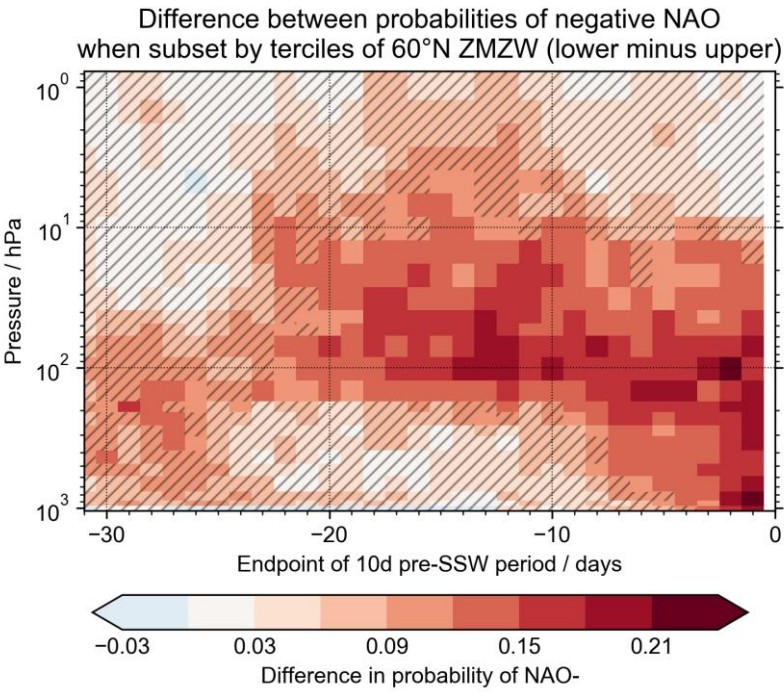


**Figure 5. Effect of zonal mean zonal wind (ZMZW) precursors on the NAO response to SSWs. The shading shows the difference in the probability of a negative NAO response, between selecting lower and upper thirds of the ZMZW, at each pressure level and pre-SSW lead time. Points are coloured on the final day of the 10d pre-SSW period they use. Points are hatched out when the difference is statistically indistinguishable from zero.**

## 4.3 Zonal wave precursors

Here we separate our sample of SSW events according to whether they are wave-2 dominated or wave-1 dominated, based on the ratio of amplitudes of the first two zonal wavenumbers, $A_2/A_1$, in the 60° N eddy geopotential height at 50 hPa on the day of the SSW. An SSW is considered wave-2 dominated if $A_2/A_1 > 1$, and wave 1 dominated if $A_2/A_1 < 1$. The frequency of SSWs that are wave-2 dominated, and the difference in 30d mean NAO responses between wave-2 and wave-1 dominated SSWs, are shown in Figure 6. The model has 140 out of 545 SSWs that are wave-2 dominated (26%, with a 95% confidence interval of 22%–30%). The proportions in the observed data are very similar, with 13 out of 34 SSWs (38%) being wave-2 dominated. The central 95% range of the model resamples is 13% to 40%, which we take to be a measure of the uncertainty in the observed frequency. This again illustrates that the model is statistically indistinguishable from the observations in this regard, and emphasizes that the observations are highly uncertain.

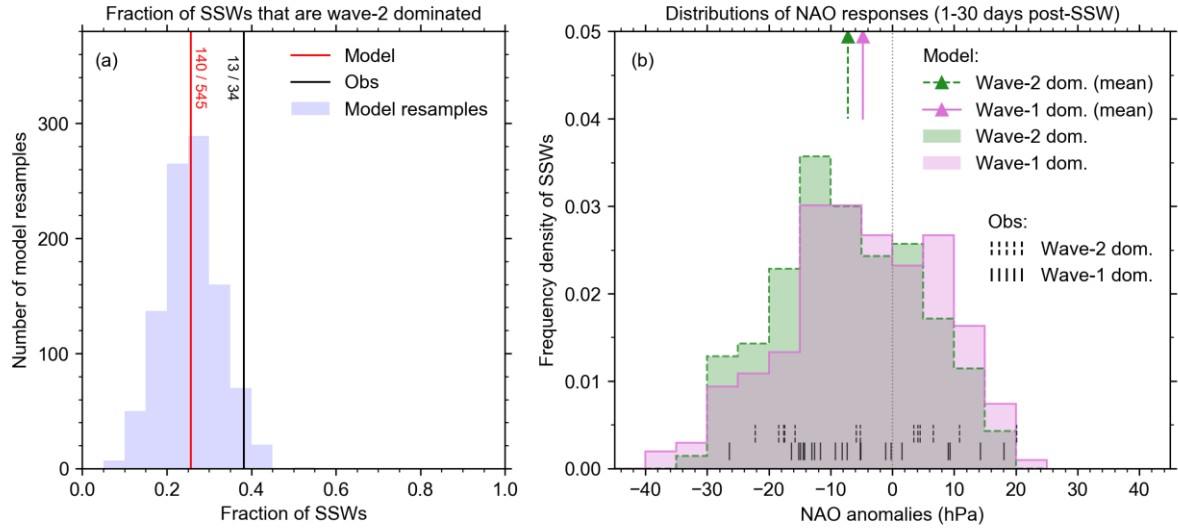


**Figure 6. Frequency of wave-2 dominated SSWs, and their NAO responses. Panel (a) shows the frequency of wave-2 dominated SSWs in the model and observations (vertical lines), and the distribution in the model resamples (histogram), in a similar format to Figure 1(b). Panel (b) shows the normalised distribution of NAO responses across the wave-2 and wave-1 dominated SSWs (as labelled), in a similar format to Figure 1(c). The distributions in the model are shown as histograms, and observations are shown as**

**vertical tick marks near the base of the plot, vertically separated for clarity. The mean responses in the model are marked with coloured arrows at the top of the plot.**

We have also tested using the wave amplitudes in the eddy geopotential height at 10 hPa instead of 50 hPa. However, as shorter wavenumbers are less able to penetrate to greater heights, following the Charney–Drazin theorem (Charney and Drazin, 1961), there are systematically fewer wave-2 dominated events by this definition: 82 out of 545 events in the model (15%), and

correspondingly 9 out of 34 in the observations (26%). These fractions in the model and observations are again statistically indistinguishable, but rather than limiting ourselves by these smaller numbers, we prefer to use the 50 hPa level to give a better representation of wave-2 prominence in the stratosphere.

Figure 6(b) shows that the 30d mean NAO responses to both wave-2 and wave-1 dominated SSWs cover a similar range to each other. The wave-2 dominated SSWs have proportionally more negative NAO responses, and fewer positive NAO

responses, than the wave-1 dominated SSWs. However, the probabilities of a negative NAO response in the wave-1 and wave-2 dominated cases (63% and 72% respectively in the model) are not significantly different.

The correlation of the wave amplitude ratio (in terms of $\log_{10} A_2/A_1$, as the ratio itself is highly skewed) with the post-SSW NAO is -0.07, which is also not significant. However, the mean NAO responses in the model *are* significantly different to each other. These are shown in Figure 6(b), and can also be seen in a wider context in Figure 7, which shows maps of the composite

mean PMSL response to wave-1 and wave-2 dominated SSWs: There is a clear negative NAO response in both cases, but the response in the wave-2 dominated case is stronger. This agrees with previous results, e.g. Seviour et al. (2013, 2016). Overall, although we can discern an impact of selecting wave-2 vs wave-1 dominated SSWs on the NAO response *on average* over the

following 30 days, this is much weaker than the other precursors that we have investigated: the wavenumber characteristics of an SSW are not a strong predictor of its NAO response over the next month.

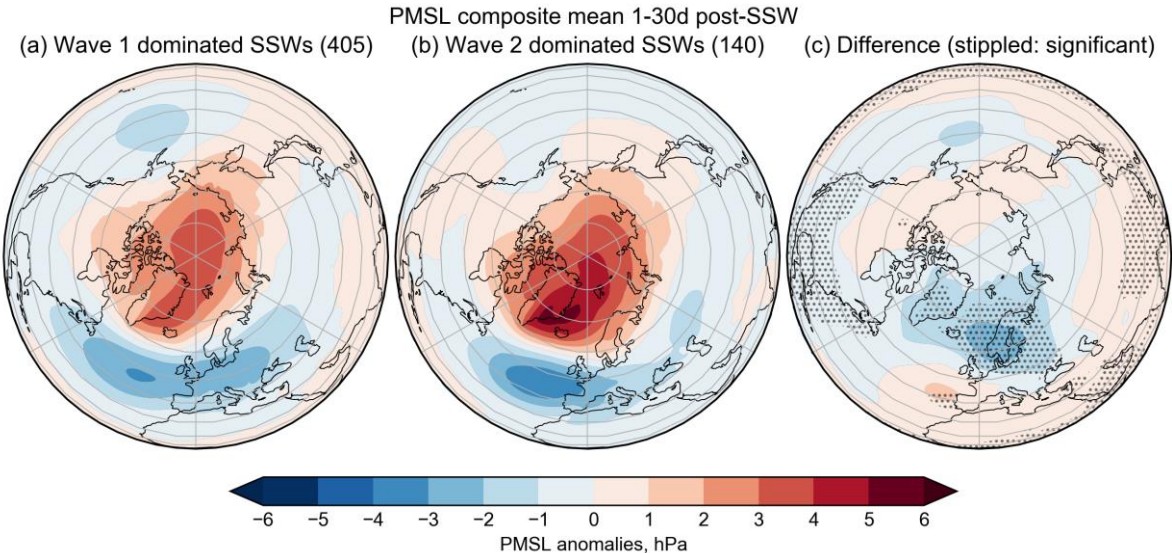

PMSL composite mean 1-30d post-SSW
(a) Wave 1 dominated SSWs (405)   (b) Wave 2 dominated SSWs (140)   (c) Difference (stippled: significant)


**Figure 7. Surface response to wave-1 and wave-2 dominated SSWs. The left (a) and centre (b) panels show the composite mean PMSL anomalies 30d post-SSW, for the two sets of SSW events. The right-hand panel (c) shows the difference (wave-1 dominated cases minus wave-2 dominated cases), with stippling indicating where the differences are statistically significant.**

Several studies have suggested that an important difference between "split" and "displacement" SSW events is the speed of the downward response, with events where the vortex splits resulting in a more rapid response from the stratosphere to the surface, and displacement events influencing the surface over longer periods (Hall et al., 2021; Mitchell et al., 2013; Seviour et al., 2013, 2016; White et al., 2019). In Figure 8, we have separated the response to wave-1 and wave-2 dominated SSWs into the first 10 days, and the subsequent days 11–30. This shows that there is a clear difference between the responses to wave-1 and wave-2 dominated SSWs immediately after the event (Figure 8a, b, c). The wave-2 cases, broadly corresponding to splits, exhibit significant differences in the sign of the composite mean PMSL anomalies in North America and Scandinavia, and a much stronger negative NAO pattern. In contrast, the later responses for the wave-1 and wave-2 dominated cases are very similar to each other (Figure 8e, f, g). The relatively small but significant differences we saw between the wave-1 and wave-2 cases for the full 30d mean response (Figure 7) is clearly a combination of the strong rapid response, and the longer-term common response.

The underlying distributions of NAO responses from individual events (Figure 8d, h) help the interpretation of these composite mean responses: For both the early and later response periods, the distributions for both wave-1 and wave-2 dominated SSWs remain very broad. The short responses show that wave-2 dominated cases have an excess of strongly negative NAO

anomalies, while the wave-1 dominated cases have an excess of positive NAO responses. This emphasizes that these results can only be interpreted probabilistically when considering future events, despite the clear differences in the composite means.

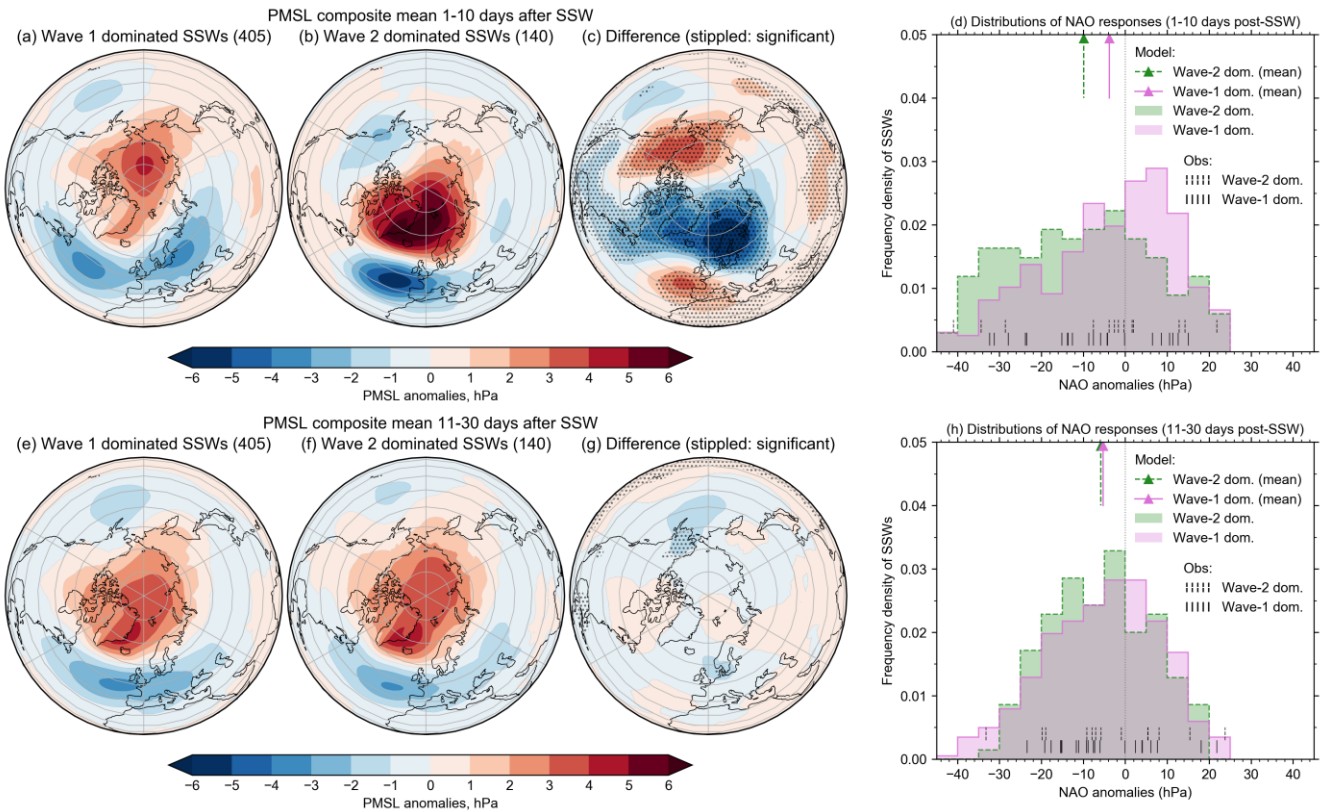

Figure 8. Surface responses to wave-1 and wave-2 dominated SSWs, for the first 10 days (top row, panels a–d) and days 11–30 (bottom row, panels e–h) after the SSW. The maps (panels a–c and e–g) show the composite mean PMSL anomalies for wave-1 and wave-2 dominated cases, and the difference between the two, as in Figure 7. The histograms (panels d and h) show the distributions of NAO anomalies over wave-1 and wave-2 dominated SSWs, as in Figure 6b.

## 5 Discussion and summary

We have shown that our ensemble of initialised climate model simulations is statistically indistinguishable from the observations in terms of the frequency of SSWs, the distribution of their NAO responses, and the proportions that are wave-1/wave-2 dominated. Other recent papers have had similar success in using initialised models, on the subseasonal (Spaeth and Birner, 2022) and seasonal (Kolstad et al., 2022; Monnin et al., 2022) timescales. Our results are in agreement with other studies in this area: for example the often-quoted frequency of about 6 SSWs per decade (e.g. Bancalá et al., 2012; Charlton and Polvani, 2007; White et al., 2019) easily falls within the range from our model resamples. Similarly our fraction of SSWs with negative NAO responses is in broad agreement with other studies quantifying the chances of surface impacts following SSWs (e.g. Domeisen, 2019; Karpechko et al., 2017; Sigmond et al., 2013), although studies examining downward-

propagating signals from SSWs in general have reported a wide range of frequencies, depending on the definitions and data

sets used (e.g. Jucker, 2016; Karpechko et al., 2017; Runde et al., 2016; White et al., 2019).

Our results have enabled us to determine conditions prior to an SSW that have a statistically significant effect on the probability of subsequent negative NAO conditions. The size of our ensemble, and its agreement with the observations, has meant that we have been able to do this more reliably and robustly than if we had based our assessments on the observations alone.

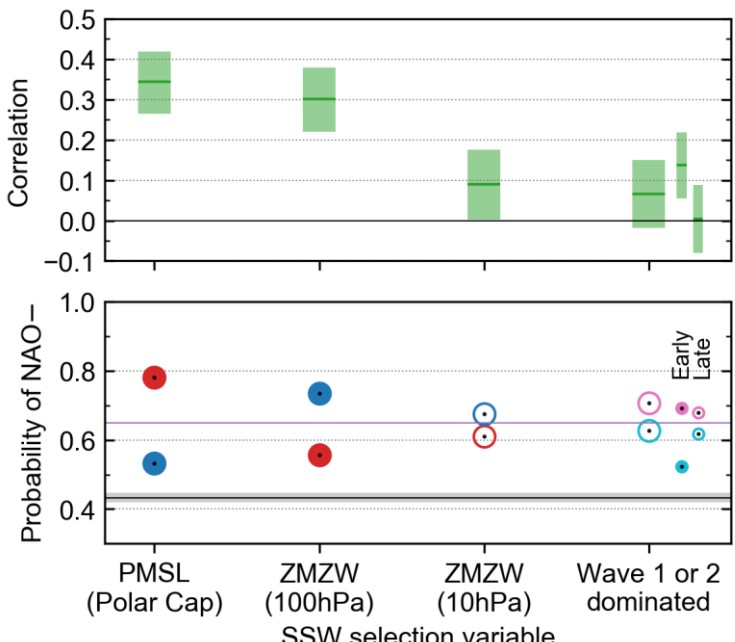

**Figure 9. Summary of the impact of different factors on the 30d mean NAO response to SSWs. Four key results are highlighted: the best-performing 10d pre-SSW PMSL region, the 10d pre-SSW 60° N zonal mean zonal wind at two different heights, and whether the SSW is wave-1 or wave-2 dominated at 50 hPa (in terms of $\log_{10} A_2/A_1$). We additionally show the impact of wave-1 or wave-2 domination on the NAO in the first 10d post-SSW (labelled "Early"), and days 11-20 ("Late"). Top: Correlation between each factor and the post-SSW NAO anomaly, with 95% confidence intervals. Correlations for the PMSL and wavenumber factors have been**
**inverted for ease of comparison. Bottom: Changes in the probability of a negative NAO, after selecting upper (red) or lower (blue) thirds of each precursor, or if the SSW is wave-2 (pink) or wave-1 (cyan) dominated. Filled circles indicate that the two probabilities are significantly different to each other. The overall rate of negative NAO responses to SSWs (as in Figure 1d) is marked with a horizontal purple line. The black line with 95% confidence interval shading gives the climatological frequency of negative NAO conditions in random 30d periods that start in DJF.**

Figure 9 shows a summary of our key results, in terms of correlations with the NAO response, and shifts in the probability of negative NAO responses, under different conditions. We have been able to rule out some features as being significant or strong determinants of the 30d-mean NAO response to an SSW: both the magnitude of the prior NAO state, and the strength of the 60° N zonal mean zonal wind at 10 hPa before the SSW, appear to have little bearing on the subsequent mean NAO state. And although we have confirmed that the NAO response in the 10d immediately following an SSW is stronger on average for wave-

2 dominated SSWs than wave-1 events, this is still a relatively weak effect compared to other factors. The same conclusion holds for responses averaged over the full 30d period.

In contrast, the PMSL anomalies over the polar cap, and the ZMZW around 100 hPa, both averaged 1-10d pre-SSW, have a significant impact on the likelihood of subsequent negative NAO conditions: they can change the probability from the baseline value of about 2/3, down to around 55% or up to around 75%, when selecting by terciles of the precursor. However, even the

reduced probabilities are greater than the climatological probability of 30-day negative NAO conditions (43%), so the presence of an SSW always increases the NAO-negative probability even given these modulating factors. The strongest correlations we have identified with the post-SSW NAO are around 0.3. With our large sample size, these are statistically significant, but it is important to emphasize that they are still relatively weak correlations.

The observed correlations (not shown) are all consistent with the model values, as the small sample size results in much wider

confidence intervals. However, the significance tests also agree with the model results. For the Polar Cap PMSL precursor for example, the observed correlation between that and post-SSW NAO is -0.59, with confidence intervals of -0.78 to -0.31, covering the model value of -0.34. For the correlation of the wave amplitude ratio with the post-SSW NAO, the observed value (0.22) is nominally the opposite sign to the model result (-0.07), but again the wide confidence intervals of -0.13 to +0.52 easily cover zero and the model value; and the model correlation is also not statistically significant.

The precursors of impactful SSWs that we have identified are in broad agreement with other studies. For example, Karpechko et al. (2017) and Oehrlein et al. (2021) showed that SSWs with stronger anomalies in the lower stratosphere are more likely to have a surface impact, and that the state of the polar vortex at 10 hPa has very little relationship with the NAO response, matching our results.

Anomalously high pressure over Eurasia and/or the Polar Cap has been seen prior to SSWs in many studies (e.g. Cohen and

Jones, 2011; Kolstad et al., 2010; Kolstad and Charlton-Perez, 2011; Mitchell et al., 2013; Seviour et al., 2013; White et al., 2019), similar to the area we identified as a precursor increasing the likelihood of a negative NAO response (Figure 4). However, differences in analysis methods and data sets, including different approaches to constructing model ensembles, make it difficult to compare directly.

Many studies have investigated the link between different vortex breakdown modes (splits versus displacements) and the

subsequent impacts. Comparison is again made more difficult by the diversity of different analysis methods revealing different results (e.g. Maycock and Hitchcock, 2015; Mitchell et al., 2013). We have used the dominant wavenumbers in the 50 hPa eddy geopotential height to show that wave-2 dominated events have a much stronger negative NAO response in the first 10 days than wave-1 events; but also that the response in the subsequent 20 days is very similar regardless of wavenumber. This has previously been seen in observation-based studies (e.g. Hall et al., 2021; with the caveat of the limited sample size), and

in dedicated modelling studies (White et al., 2021); our results show this for the first time in model data from initialised hindcast ensembles. However, it is important to emphasize that the composite mean responses are based on very broad

distributions over individual SSW events, demonstrating the necessity of using large ensembles in such studies, in agreement with Maycock and Hitchcock, (2015).

The impact of unpredictable internal variability in masking the potential impact of precursors on the surface response is also
frequently noted in the literature (e.g. Hitchcock and Simpson, 2014; Oehrlein et al., 2021; White et al., 2019), and this is reflected in the low correlations we have seen.

While we have focused on precursors of surface impacts in the pre-SSW PMSL and ZMZW fields, there are many other features of the climate that affect the behaviour and evolution of SSWs, and hence could affect the likelihood of potentially extreme surface impacts. It has been long known that the stratospheric polar vortex is weaker, and SSWs more frequent, when
the quasi-biennial oscillation (QBO) in the tropical stratosphere is in its easterly phase (e.g. Anstey and Shepherd, 2014 and references therein). The El Niño–Southern Oscillation (ENSO) affects the stratosphere, and El Niño events have been linked to an increased likelihood of SSWs (e.g. Domeisen et al., 2019 and references therein). Phases 6 and 7 of the Madden–Julian Oscillation (MJO), i.e. enhanced convection in the tropical west Pacific, have also been linked to SSWs (Schwartz and Garfinkel, 2017; Stan et al., 2022), both in terms of helping to trigger the event in the first place, and in making surface impacts
such as a negative NAO or AO pattern more likely. Although the effects of these different driving phenomena are difficult to disentangle, there have been some promising results (e.g. Liu et al., 2014; Ma et al., 2020). As with other SSW-related studies, the small observational sample is a limiting factor when drawing robust conclusions; but it also suggests that this area would be an interesting target for further research using the hindcast-based approach we have demonstrated here.

We hope that our findings, based on features of the surface pressure and polar vortex winds, will help to clarify forecast
assessments based on the state of the climate system prior to SSW events. Although precursors exist, we emphasize that they only have a modest influence on the probability of an SSW being followed by a negative NAO and its attendant impacts.

**Data availability**

ERA5 data was downloaded from the Copernicus Climate Change Service (C3S) Climate Data Store; in particular, the hourly data from 1979 at the surface and on pressure levels (Hersbach et al., 2018a, 2018b), and the corresponding data from the
preliminary back extension for 1950–1978 (Bell et al., 2020a, 2020b). Although we used GloSea5 hindcast data that was available internally at the Met Office, much of that data is also available from the C3S Climate Data Store, at https://doi.org/10.24381/cds.181d637e (surface data) and https://doi.org/10.24381/cds.50ed0a73 (data on pressure levels); system codes 12 and 13 correspond to the hindcast runs used here.

## Author contribution

PB conducted the analysis and wrote the original draft. AS, SH and HT supervised and contributed to the interpretation of the results. All authors contributed to the writing, reviewing and editing of the manuscript.

## Competing interests

The authors declare that they have no conflict of interest.

## Acknowledgements

PB would like the thank Lesley Gray and Mark Baldwin for helpful discussions during the early stages of this study. This work and its contributors (PB, AS, SH) were supported by the UK–China Research & Innovation Partnership Fund through the Met Office Climate Science for Service Partnership (CSSP) China as part of the Newton Fund. SH and HT were supported by the Met Office Hadley Centre Climate Programme funded by BEIS. The results contain modified Copernicus Climate Change Service information 2022. Neither the European Commission nor ECMWF is responsible for any use that may be made of the

Copernicus information or data it contains.

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
