# Peer review of "Using large ensembles to quantify the impact of sudden stratospheric warmings and their precursors on the North Atlantic Oscillation"

_Weather and Climate Dynamics, 2022_

## Referee Comment (RC1)

Review of "Using large ensembles to quantify the impact of sudden stratospheric warmings on the North Atlantic Oscillation" by Bett et al.

By Erik W. Kolstad

First, I'd like to congratulate the authors on a very readable and scientifically sound paper. I enjoyed reading it, and I think it will be an important contribution to the field. It is very useful to see that the observed SSW frequency and post-SSW NAO signature (in reanalyses) is within the range of model's (natural) variability, and it was interesting to read about the (lack of) precursors. The two-thirds frequency of negative NAO after SSWs seems to stand up to scrutiny. It's also encouraging to see that the downward progression of anomalies, introduced about 20 years ago, still holds. Further, I'm a fan of your method (using large ensembles for climate variability studies), and I'm glad to see that it seems to be gaining traction, with several recent or upcoming papers.

Here are some mostly minor issues that should be addressed before the paper can be accepted.

In the introduction, you only mention the UNSEEN paper when you discuss previous studies. I think you should also mention some of the many other, including earlier, papers which have used a similar method (e.g., van den Brink et al., 2004, 2005; Breivik et al., 2013; Weaver et al., 2014; Chen & Kumar, 2017; Kent et al., 2017; Kelder et al., 2020; Wang et al., 2020; Spaeth & Birner, 2021; Brunner & Slater, 2022; Monnin et al., 2022).

I'm puzzled that you used ERA-Interim and not ERA5. I guess this is the reason that you stopped in 2018/19, whereas with ERA5 you could have used the last few winters as well. This should be justified. It's hard to see why you made this choice.

I think you should comment on the periods, which don't overlap exactly (1979/80 to 2018/19 for ERA-Interim and 1993/94 to 2015/16 for the hindcasts). There was along lull in SSWs in the 1990s, and for the first part of this period you don't have hindcast data. How many SSWs are there in ERA-Interim per year between 1979/80 and 1992/93, and from 2016/17 to 2018/19, compared to the frequency during the period for which the data overlap, and how might this influence your results?

What about detrending? You don't consider temperature, and I guess the PMSL and GPH trends might be negligible, but does it merit at least one sentence (i.e., why you don't detrend the data)?

I also have an issue with your definition of the NAO, although I know that some of you have used a similar index several times before. It seems strange not to standardize the northern and southern regions separately before you take the south-north difference. The variance in the northern region is higher than in the southern region and probably dominates your NAO index.

Figure 1: I struggled a bit to understand what was shown here. I think you should explain the fraction in panel a. The way I understand it this is the number of winters with at least one

SSW out of the 40 resampled ones, and then the count on the y-axis is the number of resampled time series in each bin (of 0.05 width). Please explain more thoroughly, so that the reader doesn't have to guess what the figure shows. Once you understand panel a, panel b is easier. Panel c and d though, are tougher. What I *think* it means is as follows. Panel c shows the 30d NAO anomaly across all the 545 SSWs in the hindcast winters, independent of resampling. In Panel d, you've first computed the mean SSW anomaly across all the 40 winters in each of the 1000 resamples, and then you show the distribution of these 1000 mean values. Please explain more thoroughly. (You should also consider using a dashed line for either the black or the red vertical line to avoid black and white and color-blindness issues.)

Harking back to the lack of references to similar papers in the introduction, after you do cite some of them, perhaps you should also discuss how your results agree or disagree with their results?

Other minor issues:

1. L74: Define "SPV".
2. L75: Define "PMSL".
3. L114: What does "standard deviation" mean here? Window?
4. Are you comfortable with using "tercile" do describe the data which is separated by the terciles? Strictly speaking, the "tercile" is the 1/3 quantile itself. I'd use "lower third" instead of "lower tercile", but this is probably a matter of taste.
5. L324: Replace "climate" with "conditions"?
6. L340: Would it be better to use "determinant" instead of "determiner"?
7. L414: Something went wrong with the dash in Andrew Charlton-Perez's name here.

**References**

Breivik, Ø., Aarnes, O. J., Bidlot, J.-R., Carrasco, A., & Saetra, Ø. (2013). Wave Extremes in the Northeast Atlantic from Ensemble Forecasts. *Journal of Climate, 26*(19), 7525-7540. https://doi.org/10.1175/JCLI-D-12-00738.1

Brunner, M. I., & Slater, L. J. (2022). Extreme floods in Europe: going beyond observations using reforecast ensemble pooling. *Hydrol. Earth Syst. Sci., 26*(2), 469-482. https://doi.org/10.5194/hess-26-469-2022

Chen, M., & Kumar, A. (2017). The utility of seasonal hindcast database for the analysis of climate variability: an example. *Climate Dynamics, 48*(1), 265-279. https://doi.org/10.1007/s00382-016-3073-z

Kelder, T., Müller, M., Slater, L. J., Marjoribanks, T. I., Wilby, R. L., Prudhomme, C., . . . Nipen, T. (2020). Using UNSEEN trends to detect decadal changes in 100-year precipitation extremes. *npj Climate and Atmospheric Science, 3*(1), 47. https://doi.org/10.1038/s41612-020-00149-4

Kent, C., Pope, E., Thompson, V., Lewis, K., Scaife, A. A., & Dunstone, N. (2017). Using climate model simulations to assess the current climate risk to maize production. *Environmental Research Letters, 12*(5), 054012. https://doi.org/10.1088/1748-9326/aa6cb9

Monnin, E., Kretschmer, M., & Polichtchouk, I. (2022). The role of the timing of sudden stratospheric warmings for precipitation and temperature anomalies in Europe. *International Journal of Climatology, 42*(6), 3448-3462. https://doi.org/https://doi.org/10.1002/joc.7426

Spaeth, J., & Birner, T. (2021). Stratospheric Modulation of Arctic Oscillation Extremes as Represented by Extended-Range Ensemble Forecasts. *Weather Clim. Dynam. Discuss., 2021*, 1-25. https://doi.org/10.5194/wcd-2021-77

van den Brink, H. W., Können, G. P., Opsteegh, J. D., van Oldenborgh, G. J., & Burgers, G. (2004). Improving 104-year surge level estimates using data of the ECMWF seasonal prediction system. *Geophysical Research Letters, 31*(17). https://doi.org/https://doi.org/10.1029/2004GL020610

van den Brink, H. W., Können, G. P., Opsteegh, J. D., van Oldenborgh, G. J., & Burgers, G. (2005). Estimating return periods of extreme events from ECMWF seasonal forecast ensembles. *International Journal of Climatology, 25*(10), 1345-1354. https://doi.org/https://doi.org/10.1002/joc.1155

Wang, L., Hardiman, S. C., Bett, P. E., Comer, R. E., Kent, C., & Scaife, A. A. (2020). What chance of a sudden stratospheric warming in the southern hemisphere? *Environmental Research Letters, 15*(10), 104038. https://doi.org/10.1088/1748-9326/aba8c1

Weaver, S. J., Kumar, A., & Chen, M. (2014). Recent increases in extreme temperature occurrence over land. *Geophysical Research Letters, 41*(13), 4669-4675. https://doi.org/https://doi.org/10.1002/2014GL060300

---

## Author Comment (AC1)

**Response to reviewers**

We would like the thank both reviewers for their positive comments on the paper, and their helpful recommendations for its improvement.

An important change following comments from both reviewers has been to switch to using ERA5 instead of ERA-Interim. This has increased our observational sample from 40 years to 72. In addition to re-generating all results that include observations, our hindcast resamples have also been regenerated, as each resample now includes 72 winters instead of 40. While this is a big increase in the observational sample, it is still small compared to the 966 winters from the model. As a result, although many results have changed quantitatively, it has not changed our conclusions.

Both reviewers also found Figure 1 to be unclear. We have revised this (described in more detail below) so that its main points are more immediately clear to the reader.

Following a suggestion of Reviewer #2, we have performed a more detailed analysis of the response to wave-1/wave-2 dominated events, separating the early and later responses. This has yielded an important additional result, with a new figure, text, and mention in the abstract.

Finally, we have thoroughly revised the discussion and summary section, in response to suggestions from both reviewers to compare our results to more of the papers cited in the introduction.

We address specific comments from both reviewers in detail below.

**Reviewer 1**

In the introduction, you only mention the UNSEEN paper when you discuss previous studies. I think you should also mention some of the many other, including earlier, papers which have used a similar method (e.g., van den Brink et al., 2004, 2005; Breivik et al., 2013; Weaver et al., 2014; Chen & Kumar, 2017; Kent et al., 2017; Kelder et al., 2020; Wang et al., 2020; Spaeth & Birner, 2021; Brunner & Slater, 2022; Monnin et al., 2022).

Thanks for these suggestions. We have now extended this paragraph with more references, particularly highlighting cases where this approach has been applied to SSWs.

I'm puzzled that you used ERA-Interim and not ERA5. I guess this is the reason that you stopped in 2018/19, whereas with ERA5 you could have used the last few winters as well. This should be justified. It's hard to see why you made this choice.

We are now using ERA5. This has resulted in quantitative changes to many plots and values quoted in the text. Our conclusions remain unchanged, and indeed we feel are better supported by the comparison against the longer data set.

I think you should comment on the periods, which don't overlap exactly (1979/80 to 2018/19 for ERA-Interim and 1993/94 to 2015/16 for the hindcasts). There was along lull in SSWs in the 1990s, and for the first part of this period you don't have hindcast data. How many SSWs are there in ERA-Interim per year between 1979/80 and 1992/93, and from 2016/17 to 2018/19, compared to the frequency during the period for which the data overlap, and how might this influence your results?

The following plot shows the frequency of winters with at least one SSW, in consecutive 23-year periods in ERA5 (black dots, each marked at the beginning of the 23-year period). The decadal-scale variability you describe is clear.

We also show the distribution of frequencies in possible 23-winter periods in the model, based on 1000 samples, where we randomly pick one of the 42 members (with replacement) for each of the 23 years. This is shown as the red histogram (based at the start of the hindcast period, with the horizontal axis scaled arbitrarily).

The overall frequencies (as in Fig 1a) are shown as horizontal dotted lines.

[Figure]

The ERA5 line is lower than the model line, and the individual 23-year fractions tend to be lower. However, they are all within the range seen in the model histogram – i.e. *the model can reproduce the frequencies seen in the obs, despite being initialised in a different period*. We have added a footnote in the text before Figure 1, to highlight that we have tested this.

The question of whether the model distribution is significantly biased compared to the observations is answered using our obs-length resamples of the model data (the blue histogram in Fig 1a): the value from the obs is within the range of possible frequencies the model could generate from samples of the same length as the obs (or equivalently, the model is within the obs sampling uncertainty), even given its particular initialisation period.

What about detrending? You don't consider temperature, and I guess the PMSL and GPH trends might be negligible, but does it merit at least one sentence (i.e., why you don't detrend the data)?

Indeed, any trends in the relevant quantities are negligible. The plots below show DJF-means from ERA5 for zonal mean zonal wind averaged around 60°N at three different pressure levels; the geopotential at 60°N and 50 hPa; and the PMSL averaged over the polar cap, as examples.

[Figure]

[Figure]

[Figure]

The same is true in the model hindcasts (ensemble means are shown below), although the relatively short 23-year period would make it harder to separate trends from variability.

[Figure]

We have added a brief sentence to the methods section to indicate that trend removal wasn't necessary.

I also have an issue with your definition of the NAO, although I know that some of you have used a similar index several times before. It seems strange not to standardize the northern and southern regions separately before you take the south-north difference. The variance in the northern region is higher than in the southern region and probably dominates your NAO index.

Using that NAO index definition would make some small quantitative differences, but would not change the qualitative results. For example, the plot below (based on the Figure 1c) shows the distribution of 30-day-mean post-SSW NAO responses, with our NAO index on the left, and the difference-of-standardized-anomalies NAO index on the right:

[Figure]

The distributions and means are very similar regardless of the definition chosen. Using the difference of the pressure anomalies without standardizing is relevant, as it is this difference that physically drives chances in the winds, and hence the weather impacts.

Figure 1: I struggled a bit to understand what was shown here. I think you should explain the fraction in panel a. The way I understand it this is the number of winters with at least one SSW out of the 40 resampled ones, and then the count on the y-axis is the number of resampled time series in each bin (of 0.05 width). Please explain more thoroughly, so that the reader doesn't have to guess what the figure shows. Once you understand panel a, panel b is easier. Panel c and d though, are tougher. What I think it means is as follows. Panel c shows the 30d NAO anomaly across all the 545 SSWs in the hindcast winters, independent of resampling. In Panel d, you've first computed the mean SSW anomaly across all the 40 winters in each of the 1000 resamples, and then you show the distribution of these 1000 mean values. Please explain more thoroughly. (You should also consider using a dashed line for either the black or the red vertical line to avoid black and white and color-blindness issues.)

We have revised Fig 1, so that hopefully the key information is now clearer.

We have added panel titles and revised the axis labels, as well as annotated the main data points (the solid vertical red and black lines) to show how the numbers are related to each other, and link better to the discussion in the text. The blue histograms are now fainter: they add important uncertainty information to help interpret differences between the red and black lines, but are subsidiary to the main data points. The caption has also been revised.

We have tested the figure using the coblis colour-blindness simulator recommended by WCD and the figure remains readable under all conditions.

Harking back to the lack of references to similar papers in the introduction, after you do cite some of them, perhaps you should also discuss how your results agree or disagree with their results?

We have thoroughly revised the discussion and summary section, which now includes more complete coverage of the relevant references.

Other minor issues:

1. L74: Define "SPV".

Definition added on 2nd line of the Introduction.

2. L75: Define "PMSL".

Replaced with "mean sea level pressure" here, with the definition of PMSL remaining in section 2.2.

3. L114: What does "standard deviation" mean here? Window?

This refers to the standard deviation parameter of the Gaussian smoothing window. The text has been changed to clarify this.

4. Are you comfortable with using "tercile" do describe the data which is separated by the terciles? Strictly speaking, the "tercile" is the 1/3 quantile itself. I'd use "lower third" instead of "lower tercile", but this is probably a matter of taste.

Agreed, "terciles" has been changed to "thirds" where appropriate.

5. L324: Replace "climate" with "conditions"?

"features of the climate" has been replaced with "conditions".

6. L340: Would it be better to use "determinant" instead of "determiner"?

Done.

7. L414: Something went wrong with the dash in Andrew Charlton-Perez's name here.

This has been corrected (it might not show up in the tracked changes).

**Reviewer 2**

Title; The paper focusses on how the precursory state affects the subsequent tropospheric response to an SSW so as to aid in predictability of the response. I think this should be made more explicit in the title (refer to the 'predictability' or 'precursory' focus). The reason is that precursory features can only go so far in explaining the tropospheric response (and only in a probabalistic sense), and I think your correlations highlight this as the maximum correlation is ~0.3. What matters more mechanistically are the lower-stratospheric features after the onset date. Your current title generalises across both of these facets of the problem whereas I think a distinction should be made.

We have revised the title to highlight explicitly that we're considering the impact of precursors of SSWs on the subsequent NAO.

Lines 79-84; A couple of more recent studies that looked at the difference between displacement and splits are Hall et al. (2021; JGR) and White et al. (2021; JGR). Both found that the only salient differences in the surface response between the two occur at lags close to the onset date whereas differences in the surface response at later lags are statistically insignificant. Hall et al used reanalysis whereas White et al used an idealised GCM to artificially force displacement/splits.

Although we have not added these references at this point in the text, we have followed the later suggestion to investigate this issue ourselves, and these papers are cited later.

Lines 101-102; Why are you only focussing on those three initialisation dates? Why not also use initialisations from later November? Given your focus on DJFM, is that to allow a 'spin-up' of sorts?

Yes, we want to allow enough time for the model to spin up a diverse set of responses from the initialised conditions. However, we also want to ensure we include information about the climate prior to each winter, rather than just use a free-running model. Using the 25 Oct/1 Nov/9 Nov initialisations provides a balance between these requirements. A sentence of this has been added to the text.

Line 104; can you justify why you use ERA Interim rather than ERA-5?

We are now using ERA5. This has resulted in quantitative changes to many plots and values quoted in the text. Our conclusions remain unchanged, and indeed we feel are better supported by the comparison against the longer data set.

Line 109; Is it resampling with replacement?

Yes. This has now been noted in the text.

Lines 113-114; Can you clarify how you calculated the climatology for both the model hindcasts and the reanalysis? Presumably this describes the anomalies in the reanalysis, but did you use the same reanalysis climatology to calculate anomalies in the hindcasts?

The model hindcast and reanalysis will have different climatologies (different daily climatological means and standard deviations), but they are calculated in the same way for both data sets. For the reanalysis we had 40 winters from which to calculate them (now 72); for the model hindcast we had 23x42 = 966 winters. A note has been added to the text to clarify this.

Lines 120-121; Why not just include the events that occur in 1st-10th December and use the 10 days prior to that to examine the precursor stage? I wonder how many events would have occurred in that period anyway.

While this might look "cleaner" in the text, it would in practice involve performing every stage of analysis again from the start, including extracting raw hindcast data from tape archives. This is because the data files we have been working with are cut-outs covering DJFM only, and not November. We might also have to revisit our choice of initialisation dates.

We do not think it would add substantially enough to the paper to merit this additional work. The result is that slightly fewer SSWs contribute to Figure 4 (and 5): 507 from the model rather than 545, and 32 from ERA5 instead of 34 (in ERA-Interim, this was 17 rather than 19). We have added some text in the methods section and before Figure 4 to clarify this.

Line 147; Can you explain what the 'and significantly different to 50%' means? It is also present in other places, but clarifying once here would be useful.

This reflects the result of a significance test against the null hypothesis that the SSW frequency is 50%, using a standard binomial test. As it appears that the true value is close to the convenient round number of 50% (at least one SSW every other year on average), we felt it was worth testing if that was a reasonable approximation, or if we had enough data to show a significant difference to 50% (note that if we only had the observations, we would *not* have enough evidence to significantly rule out that the frequency was 50%). The end of the preceding paragraph notes that we'd use a binomial test in some cases, but we've now added a note to clarify that this is the test we're using here.

Line 172; It was not clear to me what panel (d) shows. (c) is also a bit confusing but I think it is the 30-day averaged NAO anomaly after EVERY SSW regardless of hindcast resample. Is that right?

We have revised Figure 1. Hopefully the improved labelling and annotation now makes the figure clearer.

We have added panel titles and revised the axis labels, as well as annotated the main data points (the solid vertical red and black lines) to show how the numbers are related to each other, and link better to the discussion in the text. The blue histograms are now fainter: they add important uncertainty information to help interpret differences between the red and black lines, but are subsidiary to the main data points. The caption has also been revised.

Lines 197-205; The Aleutian Low is usually a tropospheric precursor to SSWs (e.g., Garfinkel et al. 2010 whom you already cite) rather than a response to SSWs. Given the difficulty in teasing apart the response and cause around the onset date in observations, what would the PMSL look like when using an averaging window that does not include that first few days after the onset date, e.g., say, lags 5-30 or 7-30? It may well be that El-Nino is causing this Aleutian Low feature to persist, but it may also be that other factors at contributing.

Since our switch to ERA5, the obs panel in Fig 3 no longer shows anything of note in the Aleutian region, with no significant difference with the model results. We therefore feel it would add more confusion than clarity to the reader to include an analysis and discussion on this.

Figure 4; Why are the number of SSWs in this figure smaller than in figure 3?

In Figure 4 (and Figure 5) we require a 10d period before each SSW, so we are limited to SSWs occurring on or after 11ᵗʰ December. (See earlier question; this has been clarified in the text.)

Line 219; Can this correlation be read off from Figure 4?

No – this is the correlation between the obs and model NAO anomalies (0.19), but Fig 4 shows the correlations for the two individual regions that make up the NAO. The correlation of the difference between those regions and the post-SSW NAO can't be exactly calculated from the individual correlations.

Lines 231-237; There is also a clear region over Eastern Canada that should likely be mentioned. Additionally, the region over the Ural mountains seems to match quite well with the Siberian High as found by White et al. (2019) to be probabalistically important for the downward response.

In the obs map, only a very small part of the area in eastern Canada shows a significant correlation.

We had already included the reference to White et al. (2019) in the discussion section, but we have now added it to the similar series of papers following Fig 4.

Figure 5; This is a very interesting plot! In some ways it reframes a somewhat known conclusion that the tropospheric response depends most strongly on the lower-stratospheric anomalies than those higher up. This has conventionally been shown using scatter plots (e.g., Maycock and Hitchcock 2015; Karpechko et al. 2017) but yours highlights the predictability change across all heights and precursory lags.

Thanks for the positive comments!

Section 4.3 and Figures 6-7; Above I mentioned two papers (Hall et al. 2021 and White et al. 2021) that found that the difference between splits and displacements (equivalent to your wave-1 and wave-2 dominated events) was most pronounced at lags close to the onset date, within about 1-10 days or so. After that, the differences were very small and statistically indistinguishable from one another. Your 30-day window covering lags 1-30 after the onset merges these two periods together and I think it would be worthwhile to recreate these two figures for, say, lags 1-10 and 11-30 to see if indeed there are wave-1 vs wave-2 differences in the earlier period relative to the latter.

This was indeed worthwhile, yielding a very clear response in agreement with the results cited: we see a strong response in the NAO in the first 10 days for wave-2 dominated events, but extremely similar responses between wave-1 and wave-2 dominated cases for the subsequent 11–30-day period. These results are shown in the new Figure 8 and surrounding text, with additional material added in the abstract and discussion & summary.

Lines 358-360; I feel that the Xu et al. study as described disagrees with your results, no? You find the middle stratosphere to be poorly correlated with the surface response compared to the lower stratosphere.

Our use of "middle stratosphere" was confusing. Xu et al. find enhanced NAO- responses for SSWs with more weakened prior vortex over the whole depth of the stratosphere (10 hPa down to the tropopause), but an NAO+ response when only the vortex above 50 hPa is weakened. It is therefore rather difficult to compare with our work directly, and this is made more difficult by their use of only

observational data, resulting in very few events in their four categories. We have removed this discussion, but added a citation to the Xu et al. paper to the introduction.

Summary and Discussion; You do not refer back to many of the studies listed in the introduction. All do not need to be referred to here, but a good proportion should likely be.

We have thoroughly revised the discussion and summary section, which now includes more complete coverage of the relevant references.

---

## Author Response (AR2)

**Response to reviewers**

We would like to thank the reviewer for their additional comments on the paper, which we address below.

**Reviewer 1**

line 97: "assessing the risk in the southern hemisphere": the risk of what? please clarify

Text changed to read "assessing the chance of southern hemisphere sudden warmings and associated risk of extreme hot/dry conditions in austral subtropical continents".

line 197: typo: "a similar results"

Fixed.

Figure 9: it is a bit confusing how red and blue mean different things for different factors, i.e. upper or lower tercile or wave-2 vs wave-1. would it be possible to use e.g. different colors for these or to add a legend to the figure that would clarify this?

Agreed, thank you.  We have changed the colours corresponding to wave-2 vs wave-1.

line 411: "It is also therefore weak when considering the responses when averaged over the full 30d period." I think this sentence could be clarified a bit more, e.g. clarify what "it" refers to.

Text clarified.

line 457: "in dedicated modelling" change to "in dedicated modelling studies"

Changed.

line 465: would only use the word "significant" when meaning "statistically significant". if that's what is meant here then please keep it

Changed to "potentially extreme".

line 469: if you'd like to cite a study that uses initialized models for the MJO teleconnection to the stratosphere, could use this one: https://journals.ametsoc.org/view/journals/bams/103/6/BAMS-D-21-0130.1.xml

Thank you – now cited.